# Mindfulness-Based Movement Intervention to Improve Sleep Quality: A Meta-Analysis and Moderator Analysis of Randomized Clinical Trials

**DOI:** 10.3390/ijerph191610284

**Published:** 2022-08-18

**Authors:** Jiayi Yang, Yan Du, Haoran Shen, Shujie Ren, Zhiyuan Liu, Danni Zheng, Qingqing Shi, Youfa Li, Gao-Xia Wei

**Affiliations:** 1Collaborative Innovation Center of Assessment for Basic Education Quality, Beijing Normal University, Beijing 100875, China; 2School of Nursing, Health Science San Antonio, University of Texas, Austin, TX 78712, USA; 3CAS Key Laboratory of Behavioral Science, Institute of Psychology, Chinese Academy of Sciences, Beijing 101408, China; 4Department of Psychology, University of Chinese Academy of Sciences, Beijing 101408, China; 5School of Education, Beijing Sport University, Beijing 100084, China; 6School of Psychology, Beijing Sport University, Beijing 100084, China

**Keywords:** mindfulness-based movement, sleep quality, meta-analysis, randomized controlled trials

## Abstract

(1) Background: Given that the most effective dose, optimal type, and most beneficial population for improving sleep with mindfulness-based movement (MBM) remains unknown, we conducted a systematic review and meta-analysis with moderator analysis of randomized controlled trials (RCTs) to assess these effects. (2) Methods: Three electronic databases (PubMed, Web of Science, and EBSCO) were systematically searched for RCTs published through August 2021 for analysis. The risk of bias of the included studies was assessed with Review Manager 5.3, and the meta-analysis was performed in Stata 16.0. (3) Results: A meta-analysis of 61 RCTs with 2697 participants showed that MBM significantly improved sleep quality compared to controls (SMD = −0.794; 95% CI: −0.794 to −0.994, *p* < 0.001, I^2^ = 90.7%). Moderator analysis showed that a long-term MBM (SMD = −0.829; 95% CI: 0.945 to 0.712; *p* < 0.001) had a larger effect size on sleep than a short-term MBM (SMD = −0.714; 95% CI: 0.784 to 0.644; *p* < 0.001). Practicing at least twice per week (SMD = −0.793; 95% CI: −0.868 to −0.718; *p* < 0.001) was more effective compared to practicing once per week (SMD = −0.687; 95% CI: −0.804 to −0.570; *p* < 0.001). Studies with a total intervention time of more than 24 h also revealed better sleep quality improvement (SMD = −0.759; 95% CI: −0.865 to −0.653; *p* < 0.001). In addition, the healthy population and older adults gained more from MBM than the patients and younger adults. (4) Conclusions: MBM can effectively improve subjective sleep quality, and the optimal intervention dose of MBM can be utilized in future intervention studies to treat or improve sleep disturbance (MBM more than twice a week for more than three months, with a total intervention time of more than 24 h).

## 1. Introduction

Sleep disturbances have become a significant public health concern worldwide. It is a potential risk factor for a wide variety of somatic diseases and common symptoms of mental disorders [1]. The most widely recognized treatments for sleep disturbances are pharmacological therapy and psychotherapy. Apart from the direct and indirect costs of diseases, the clinical benefits of these two first-line treatments have been criticized [2]. Pharmacological therapy can produce side effects [3,4], while psychotherapy requires time-consuming input from health professionals [5]. Currently, an increasing number of studies have examined whether mindfulness practices can effectively alleviate some aspects of sleep disturbance. A meta-analysis of 18 trials showed that compared with specific active controls, mindfulness meditation significantly improved sleep quality [6]. However, considering the improved efficacy of perceived sleep quality, a recent randomized controlled trial found that the exercise group had a better effect compared with the mindfulness group and the waitlist group [7].

With the development of mindfulness practice, integrative practice combining mindfulness and other exercise regimens has been explored. Mindfulness-based movement (MBM) has received the most attention for its alternative efficacy in many psychotic disorders [8]. It is a group of integrative techniques, including yoga, Tai Chi Chuan, Pilates, Baduanjin, and Qigong. Compared with monotherapies, MBM integrates mindfulness and aerobic exercise characteristics to boost the interaction between the brain, mind, and body by performing musculoskeletal stretching, slow breathing, and body control [9]. A meta-analysis of 49 studies assessed the effects of MBM on insomnia, indicating that MBM improved perceived sleep quality and reduced insomnia severity [10]. However, an earlier review demonstrated that nearly one-fourth of studies on mind–body interventions failed to observe their beneficial effects on sleep outcomes [11]. Therefore, the results of the MBM-induced improvement in sleep quality are inconclusive. One possible explanation for this inconsistency might be the different intervention protocols performed in the included studies. For instance, Chen et al. used an intervention protocol of 70 min per session, three times per week for 24 weeks [12], which found a beneficial effect of MBM on sleep quality. In contrast, 90 min per session, twice per week for 16 weeks, failed to find significant improvements in sleep with MBM [13]. In addition, the heterogeneity of participant characteristics may lead to inconsistent results. Yeh et al. found significant improvement in sleep quality with MBM in heart failure patients with a mean age of 59 ± 14 years [14], whereas Sakuma et al. did not observe such a benefit in healthy female child-care workers with a mean age of 32.6 ± 11.5 [15]. With the increase in the number of relevant studies in recent years, there is a need to update existing MBM studies to explore the best intervention protocols and the most beneficial populations.

To address these questions, we conducted a systematic review and meta-analysis of RCTs to assess the effectiveness of MBM on sleep quality in different populations and compare the effect sizes of improved sleep quality induced by varied intervention doses of MBM, which provided insight into health promotion among non-clinical individuals and the strategy for clinical treatment of sleep disorders.

## 2. Methods

This study was performed in accordance with the Preferred Reporting Items for Systematic Reviews and Meta-Analyses (PRISMA) guidelines [16] and the Cochrane Collaboration recommendations for systematic reviews and meta-analyses [17].

### 2.1. Search Strategies

Three English-language databases (PubMed, Web of Science, and EBSCO) were systematically searched. A similar systematic review was conducted from its inception to the end of 2006. This study focused only on those from January 2007 to August 2021. Relevant terms were searched based on three search levels: (i) yoga, Tai Chi/Taiji, Pilates, Qigong/Ch’i Kung, mind-body therapy/practice or meditative movement; AND (ii) sleep or insomnia; AND (iii) randomized or randomized controlled trial. Furthermore, reference lists of other reviews and relevant studies were manually searched.

### 2.2. Study Eligibility

The lead author conducted an independent literature search to screen the titles, abstracts, and manuscripts of eligible studies. The studies were considered to be included if an article (1) used RCT design; (2) was published in a peer-reviewed journal; (3) used an intervention that was solely or mainly based on yoga, Tai Chi Chuan, Pilates, Baduanjin, and Qigong, or combined mode; (4) included at least two groups: one was the intervention group, the other was the control group (active control or waitlist group); and (5) included studies with at least one primary outcome related to sleep (the score of sleep scale or sleep physiology outcome). The excluded criteria included observational studies, case reports, controlled trials with no randomization, and review studies. 

### 2.3. Data Extraction and Management

The retrieved articles were independently reviewed and extracted by five authors (JY, SJ, ZY, QQ, DN) to select relevant articles. For the included studies, the following information was extracted: study characteristics; the characteristics of participants (sample size, gender, mean age, and health status); intervention protocol; and outcomes measured. The primary outcomes were subjective sleep quality measured by self-report questionnaires or objective sleep data measured by polysomnography (PSG). The number of participants in the intervention and control groups and the mean ± standard deviation (SD) at baseline and after the intervention were input from each study. We contacted the corresponding author via email for data requests if the data were insufficient or missing. Complete data extraction information is available in the Appendix A.

### 2.4. Assessment of Methodological Quality

The methodological quality of the selected trials was assessed using the Cochrane Risk of Bias Tool [18]. The original scale consists of 11 items, including randomness of allocation order (selection bias), allocation order concealment (selection bias), blinding of participants and staff (performance bias), blinding in outcome assessment (detection bias), incomplete outcome data (attrition bias), and selective outcome reporting (reporting bias). Given the impracticality of blinding participants and instructors during the intervention, this item was removed from the original scale, leading to a final number of 6 items in the scale. Each individual item was examined to objectively evaluate the risk of bias across trials. The criteria were categorized as high risk of bias, unclear risk of bias, or low risk of bias. Points were awarded high scores, indicating better methodological quality. 

### 2.5. Data Synthesis and Data Analysis

We performed a sensitivity analysis to examine whether individual studies disproportionately influenced the results. The trim-and-fill method, as a measurable effect on potential publication bias (asymmetry of the funnel plot), was used for estimating and adjusting pooled standardized mean differences (SMDs) based on the funnel plot. All analyses of pooled effectiveness were conducted using STATA version 16.0 [19]. The random effects method was applied to the heterogeneous population involved in this study [20]. SMD with 95% confidence intervals (CIs) was calculated as the difference in means between groups divided by the pooled standard deviation. We categorized the magnitude of the intervention effect, with SMD as 0.2–0.5 indicating small effect, 0.5–0.8 indicating moderate effect, and 0.8 and above indicating large effect [21]. 

Between-study heterogeneity was evaluated using the I^2^ statistic (low heterogeneity = 0~25%, moderate heterogeneity = 26~50%, substantial heterogeneity = 51~75%, and considerable heterogeneity = 76~100%) [17]. 

We conducted moderator analyses with meta-regressions to provide more detailed prescription recommendations for MBM intervention doses. The moderator variables were the population based on their health status; participants’ age; the type of MBM; duration of intervention; frequencies of intervention; and total duration of intervention.

## 3. Results

### 3.1. Trial Selection

Both electronic and manual searches resulted in 1574 studies (Figure 1). After removing duplicates, 1206 studies were included according to the inclusion criteria, and 64 eligible RCTs were eventually entered into the systematic analysis after stepwise screening. The detailed trial selection process of the study is illustrated in Figure 1. 

### 3.2. Study Characteristics

The study characteristics of the selected trials are summarized in Table A1 [12,22,23,24,25,26,27,28,29,30,31,32,33,34,35,36,37,38,39,40,41,42,43,44,45,46,47,48,49,50,51,52,53,54,55,56,57,58,59,60,61,62,63,64,65,66,67,68,69,70,71,72,73,74,75,76,77,78,79,80,81,82,83,84]. These studies involved 2697 participants aged from 6 to 78 years. Nineteen studies included younger adults under 45 years old, 20 studies included middle-aged adults between 45–59 years, and 18 studies included older adults above 60 years. As for health status, 19 studies examined healthy populations; subjects in 28 studies had clinically physical illnesses with significant distress and/or dysfunction (e.g., diabetics, cancer patients, hemodialysis patients, etc.), and 17 had psychiatric disorders (insomnia, depression, menopause, and anxiety). Twenty-three studies included both males and females, 11 studies included only females, and two studies included only males.

Regarding the type of MBM, 25 studies used yoga/Pilates, 18 used Tai Chi/Qigong/Baduanjin, and 18 used MBM combined with mindfulness or muscle relaxation techniques. The duration of the intervention varied widely, ranging from 1 week to 12 months, with most studies providing an 8-week intervention (*n* = 9). The intervention frequency varied from once every two weeks to three times a day, with the studies frequently including once-a-week practice (*n* = 14) or twice-a-week practice (*n* = 13). Subjective measures of sleep quality were used in all 64 articles. Six studies used PSG to assess the physiological index related to sleep quality. No adverse events related to MBM were reported in the included articles.

### 3.3. Study Quality Assessment

The risk of bias is summarized in Figure 2. According to the Cochrane Risk of Bias Tool, the overall risk of literature bias was low in 64 studies. All studies were RCTs and used sufficient randomization. After removing the studies with inconsistent baseline periods [72], there was no heterogeneity in baseline sleep indicators between the two groups (I^2^ = 19.2%, *p* = 0.062 > 0.05). 

To detect the consistency in the effect of MBM on sleep quality, a sensitivity analysis was performed to remove two studies with outrageous effect sizes based on visually asymmetrical funnel plots (SMD = 0.5; SMD = 0.46) [48,53] (Figure 3). Egger’s test indicated that the other 61 studies had low publication bias with good symmetric characteristics (*p* = 0.191 > 0.05) (Appendix A). Sensitivity analysis confirmed the stability of the results (Figure 4). 

### 3.4. Effects of MBM on Sleep Quality Measured by Scales

Due to the large heterogeneity test (I^2^ = 90.7%, *p* < 0.001), a random-effects model analysis was employed for the overall effect. For the meta-analysis with 61 studies, compared with controls, aggregated results showed significant benefit in favor of MBM on sleep quality (SMD = −0.794, 95% CI: −0.794 to −0.994, *p* < 0.001, I^2^ = 90.7%, Figure 5). 

### 3.5. Effects of MBM on Sleep Quality Measured with PSG

A total of six studies reported data on PSG to assess participants’ sleep quality. Moon et al. [76] measured salivary cortisol and blood pressure changes, and Buchanan et al. [85] had incomplete polysomnographic data, so these two articles were excluded. Four other studies reported total sleep time (TST) data. The meta-analysis found no significant difference in total sleep time between the participants receiving MBM and the controls (SMD = −0.126; 95% CI: −0.479 to 0.227; *p* = 0.485; I^2^ = 0%).

### 3.6. Moderator Analysis

To further identify the moderator effects in MBM studies, we considered the following variables: subgroup of the population (the healthy population, the patients with somatic diseases, and the patients with psychiatric illnesses), the age of participants (youth: <45 years; middle age: 45~59 years; older adults: >60 years), duration of the intervention (≤3 months, >3 months), types of MBM (yoga/Pilates, Tai Chi/Qigong/Baduanjin, MBM combined with mindfulness or muscle relaxation techniques), intervention frequencies (≤1/week; >1/week), and total intervention duration (≤24 h; >24 h). We used a random-effect model to conduct the moderator analyses. The number of studies included in each moderator analysis varied due to incomplete data. The number of studies and participants are provided in Appendix A. 

In terms of the subgroup of participants, the healthy population showed the largest effect size compared with other subgroups, although MBM intervention showed a significantly improved sleep quality in all these subgroups: healthy population (SMD = −0.899; 95% CI: −1.000 to −0.798; *p* < 0.001), the patients with somatic diseases (SMD = −0.627; 95% CI: −0.736 to −0.519; *p* < 0.001), and the patients with psychiatric illnesses (SMD = −0.618; 95% CI: −0.720 to −0.516; *p* < 0.001) (Appendix A). As shown in Appendix A, all three age groups showed significant improvements in sleep quality after the intervention. MBM had a greater benefit on sleep quality in older adults (SMD = −0.873; 95% CI: 0.985 to 0.761; *p* < 0.001) compared to younger adults (SMD = −0.618; 95% CI: 0.715 to 0.520; *p* < 0.001) and middle-aged adults (SMD = −0.691; 95% CI: 0.803 to 0.579; *p* < 0.001). 

Compared with the MBM combined with mindfulness or muscle relaxation techniques (SMD = −0.741; 95% CI: −0.849 to −0.633; *p* < 0.001) and Taijiquan intervention (SMD = −0.567; 95% CI: −0.680 to −0.455; *p* < 0.001), the intervention of yoga/Pilates (SMD = −0.808; 95% CI: −0.901 to −0.715; *p* < 0.001) had greater effects on sleep quality (Appendix A). In terms of the duration of MBM (short-term or long-term), long-term interventions (>3 months) were more effective (SMD = −0.829; 95% CI: 0.945 to 0.712; *p* < 0.001) than short-term interventions (≤3 months) (SMD = −0.714; 95% CI: 0.784 to 0.644; *p* < 0.001) in improving sleep (Appendix A). Another moderator analysis on frequency showed that ≥2 times per week (SMD = −0.793; 95% CI: −0.868 to −0.718; *p* < 0.001) had a greater impact on improving participants’ sleep than once per week (SMD = −0.687; 95% CI: −0.804 to −0.570; *p* < 0.001) (Appendix A). In a moderator analysis of the total intervention duration, a greater effect on sleep improvement was observed for total intervention time above 24 h (SMD = −0.759; 95% CI: −0.865 to −0.653; *p* < 0.001) than for total intervention duration below 24 h (SMD = −0.746; 95% CI: −0.827 to −0.665; *p* < 0.001) (Appendix A).

## 4. Discussion

This systematic review and meta-analysis with moderator analysis updated the evidence from RCTs using MBM as an intervention for sleep problems in both non-clinical populations and patients with illness/disorders. The pooled results indicate that MBM improved sleep quality, as measured by self-report scales rather than by PSG. Further, the moderator analysis demonstrated that MBM with >3 months, twice or more per week, and longer than 24 h of total intervention had larger effect sizes on sleep than other protocols. In addition, healthy people and older adults gained more from MBM than patients with physical or psychiatric disorders or younger and middle-aged individuals. Such promising results suggest that MBM is an alternative or augmentation strategy for improving sleep quality. Additionally, as reflected in our assessment of the quality of all studies in Section 3.3, the relatively high quality of the included RCTs makes our conclusions comparatively reliable. Similarly, our results are reliable according to the consistency of Egger’s test and the sensitivity analyses.

Two previous meta-analyses of cancer patients and insomnia have illustrated that the effect of MBM on alleviating sleep quality was significant [10,86], which was roughly in line with this study with moderate to large effect sizes (Section 3.4). Wang et al. found that the effect size of MBM-induced improvement of sleep quality was moderate after reviewing 49 studies, including meditation, between 2004 and 2018 [10]. A recent meta-analysis of cancer patients found that although the effect of MBMs was smaller than that of aerobic exercise, both of these interventions significantly improved sleep outcomes [86]. Similarly, our study adds to the fuller evidence of the effect of MBM on sleep quality. 

A moderator analysis was performed for the different groups. We found that MBM had a greater impact on sleep quality in healthy individuals than in clinical populations and psychiatric patients. A possible explanation might be the complications of somatic disease [87,88] or mental disorders [89] in those clinical patients. Chronic pain [90], failure of emotional regulation [91] or medication side effects [92,93], and other medical conditions are essential variables when considering the reduced effect of MBM in those patients [94]. Healthy individuals are more likely to be capable of attending to their bodies and minds during practice, resulting in a better intervention effect.

Our study suggests that older adults benefit more from MBM than other age groups. For older adults, MBM has been shown to help prevent falls [95], improve depression [96], reverse metabolic disease [97], and reduce systemic inflammation [98,99]. A single-site qualitative study found that some older adults lose the stress of work and activity after retirement, so MBM may provide them with a new way to increase physical activity, reduce anxiety, decrease inflammatory factors [100], and consequently improve sleep quality. Unexpectedly, we found that yoga/Pilates practice had a greater effect on sleep compared to other MBMs. Although we cannot provide the reason for clarifying this finding so far, it is tempting to speculate about the potential reason, which might be due to the involvement of more females in Yoga/Pilates intervention. Therefore, gender differences should be fully considered when examining the association between MBM and sleep quality in future investigations. 

In this study, we found that the improvement in sleep with MBM was most pronounced when the total intervention time exceeded 24 h, at least twice a week, and for more than three months. Our findings on the dose-response are consistent with the previous meta-analysis reporting that improvements in sleep quality were greater when the intervention’s frequency, duration, and total duration were achieved [10]. Several studies have shown that only MBM intervention longer than three months leads to significant improvements in metabolic profiles [101,102], proinflammatory cytokine [103], and mental health [104,105,106]. Furthermore, the inverted U-shaped hypothesis proposes that moderate-intensity exercise has the most significant cognitive benefits [107,108]. Thus, a minimum dose of physical exercise is crucial to gain measurable benefits. Regarding exercise intensity, MBM belongs to practice with low-to-moderate intensity [109], so higher frequency and intervention length may predict better overall intervention outcomes. 

Considering the mechanisms underlying the effect of MBM on sleep quality, several key factors, including appropriate emotional regulation, maintaining homeostatic effect in the autonomic nervous system, and altering the inflammatory process, might contribute to improving sleep quality [110]. 

First, the improvement in sleep quality induced by MBM could be closely associated with the role of alleviating negative moods. Self-reported sleep problems and emotional disorders mutually affect each other. Oh et al. found that 21.7% of patients with insomnia had anxiety disorders, 7.2% had depression, and 18.6% had both anxiety and depression [111]. In turn, 74% of patients with anxiety disorders reported sleep disturbances [112]. Although the exact cause and effect between emotional disorders and poor sleep quality remain unknown, it has been shown that MBM has the potential to effectively treat both major depressive disorder [113,114,115] and sleep problems [81,103]. The alleviated negative moods often accompany improved sleep quality after practicing MBM [81,116,117]. A randomized controlled trial of a mind-body-spirit intervention found that changes in depressive symptoms mediated treatment effects on nighttime sleep quality and daytime functioning [116]. Additionally, a growing body of brain imaging studies has shown that MBM induces structural and functional changes in key brain regions associated with emotional regulation [118]. For instance, Tai Chi Chuan practitioners exhibited greater cortical thickness in the middle frontal sulcus [119] and elderly yoga practitioners showed significantly cortical thickness in the left prefrontal lobe [120]. The prefrontal cortex, as an “immune system of the mind”, is a flexible hub for regulating an individual’s negative emotions. MBM will likely implement its function in mediating sleep via neural pathways between the prefrontal cortex and limbic system related to emotional processing [118]. Therefore, MBM could possibly improve sleep quality by reorganizing the anatomical structures or functions of emotional circuits. 

Second, another physiological factor may explain why MBM could positively influence sleep quality in almost all subgroups. It is known that the autonomic nervous system plays a fundamental role in maintaining physiological function and body homeostasis [121]. Many studies have used non-invasive techniques to assess the changes in heart rate and heart rate variability (HRV) in autonomic nervous system activity after performing MBM practice. For instance, a study used traditional electrocardiogram recordings to examine Tai Chi practice-induced changes in HRV, which indicated that Tai Chi could improve vagal activity and the balance between sympathetic and parasympathetic activity during the relaxation state [122]. Another meta-analysis, including 19 medium-to-high quality RCTs, also demonstrated that mind–body practice could significantly benefit HRV parameters and improve sympathetic–vagal balance [123]. Higher HRV during wakefulness is associated with higher sleep efficiency and better sleep quality [124,125,126]. Therefore, the action of HRV alteration may be an important mechanism by which MBM ameliorates the symptoms of insomnia or poor sleep quality. Moreover, studies have shown that high sleep onset latency and poor sleep quality are associated with higher resting heart rates [127], which indicates that heart rate is a crucial physiological marker that reflects sleep quality. Regular moderate MBM strengthens the heart muscle and increases the oxygenation efficiency of the heart [128,129], which can improve sleep quality by lowering an individual’s resting heart rate. 

Third, a possible explanation might also be associated with the altered inflammatory processes. Many studies have shown that inflammatory biomarkers, such as Nuclear Factor-kappa B (NF-κB) play a critical role in activating inflammation in chronic insomnia [130,131,132]. Acute sleep loss can induce a rapid increase in the activation of the transcription factor NF-κB in peripheral blood mononuclear cells [133,134]. Recent studies examining changes in gene expression induced by meditation and related MBM have found that these practices are associated with downregulation of the NF-κB pathway [31,135,136], suggesting that MBM practices may lead to a reduced risk of inflammation-related diseases, including insomnia [137]. Future interdisciplinary studies are warranted to further investigate the interaction between the molecular and psychological changes associated with MBM.

Our study has some limitations. First, we searched only three databases for studies published in English, which may limit the generalizability of our results to some extent. We will consider ongoing updates and replenishment of the databases in the future. Second, only a limited number of studies and a relatively small sample size provided physiological measures of sleep duration. Although a significant effect was observed on subjective sleep outcomes, we were unable to detect improved sleep quality measured by physiological parameters. Further investigation of subjective and physiological sleep is needed in future studies. Third, our review did not consider gender differences in the meta-analysis because 25 studies did not report the male-to-female ratio of participants, and 23 studies had participants of mixed gender. In the future, more comparative RCTs explicitly targeting the effect size of different gender groups are needed to determine the role of gender in the effect of MBM on sleep quality. 

## 5. Conclusions

This meta-review suggests that MBM could be applied as a complementary or supplementary therapy to improve sleep quality. The optimized dose of the intervention (twice a week for more than three months, with a total intervention time of more than 24 h) is recommended in future interventions, both in clinical patients and in healthy individuals. Future research should address more objective measurements of the methodological aspects and individual difference issues.

## Figures and Tables

**Figure 1 ijerph-19-10284-f001:**
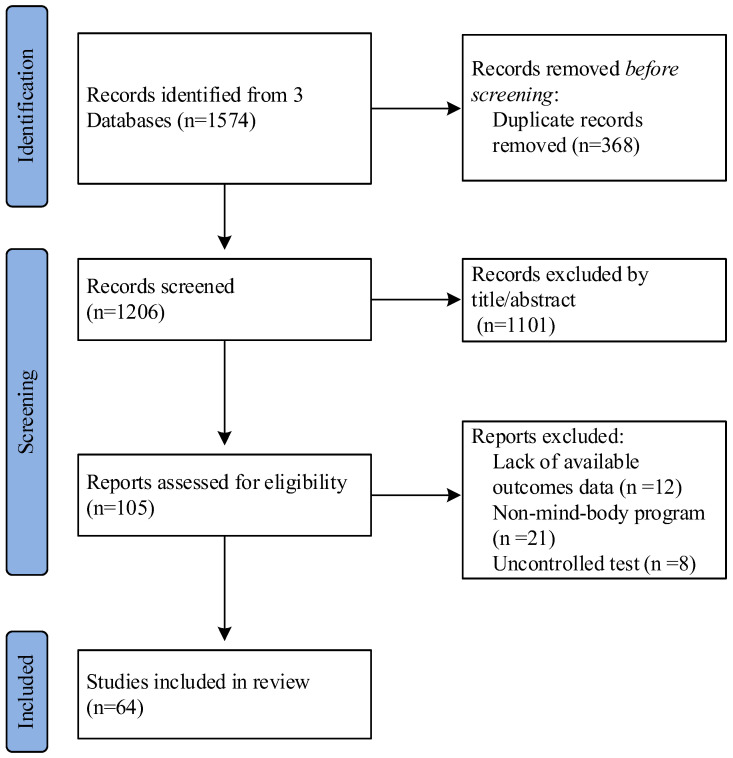
The detailed process of trial selection.

**Figure 2 ijerph-19-10284-f002:**
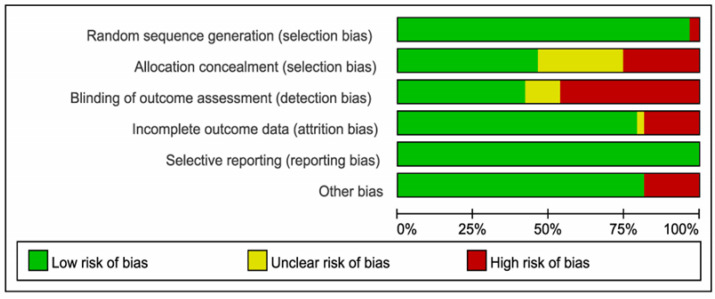
Risks of Bias within Studies.

**Figure 3 ijerph-19-10284-f003:**
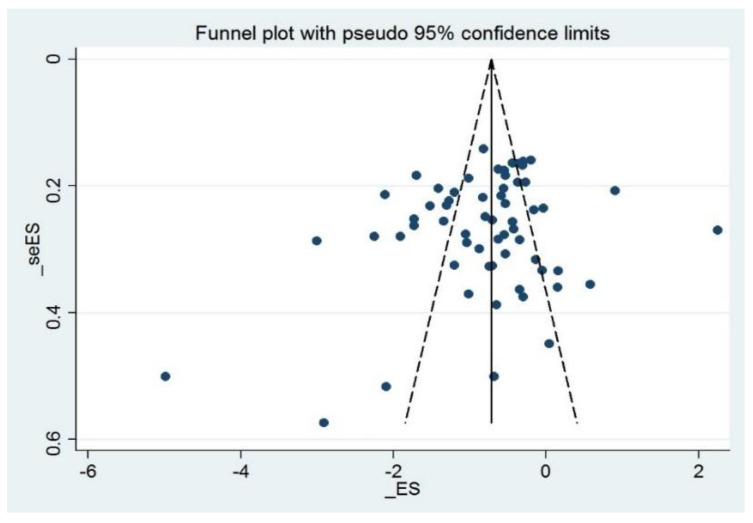
Funnel plot with pseudo 95% confidence limits.

**Figure 4 ijerph-19-10284-f004:**
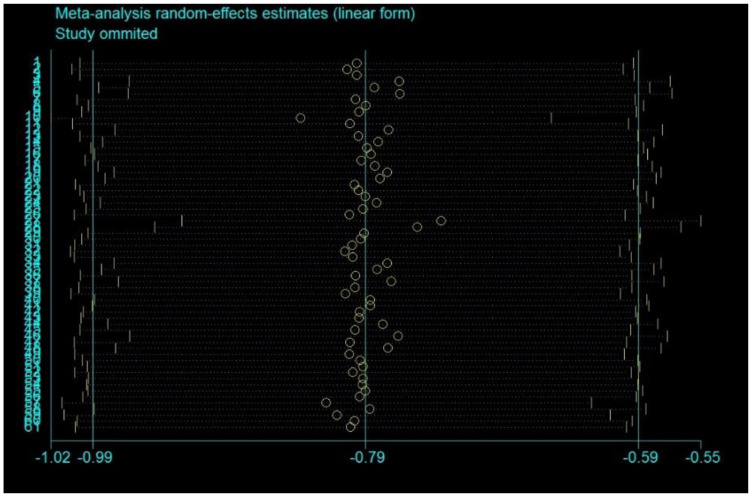
Meta-analysis random effects estimates (linear form). Note: The vertical coordinate is the included studies, and the horizontal coordinate is the confidence interval limit.

**Figure 5 ijerph-19-10284-f005:**
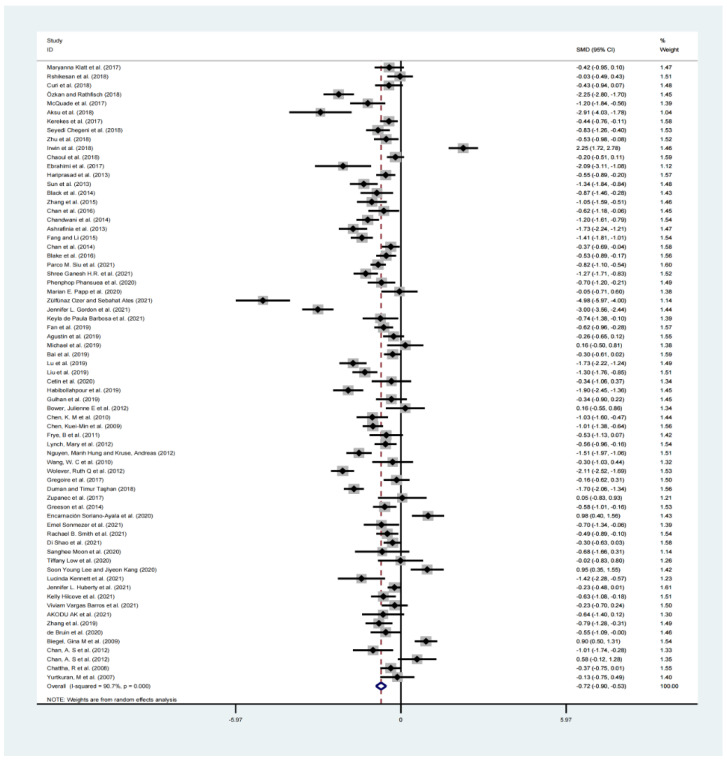
Forest plots of subjective outcomes in the overall analysis [12,22,23,24,25,26,27,28,29,30,31,32,33,34,35,36,37,38,39,40,41,42,43,44,45,46,47,48,49,50,51,52,53,54,55,56,57,58,59,60,61,62,63,64,65,66,67,68,69,70,71,72,73,74,75,76,77,78,79,80,81,82,83,84].

## Data Availability

The study did not report any data.

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
