# Peer review of "Mindfulness-Based Movement Intervention to Improve Sleep Quality: A Meta-Analysis and Moderator Analysis of Randomized Clinical Trials"

_ijerph, 2022, doi:10.3390/ijerph191610284_

Round 1
Reviewer 1 Report
This paper presents a meta-analysis on effectiveness of MBM procedures from datasets available in the existing literature.
The authors overall have done a good job. I find the paper well-written and I do think that the paper should be published.
My only comment is that the results and findings should be presented in a more reader-friendly way. I would suggest the big table in subsection 3.2 be put at the end as an appendix.
More important, the discussion in section 4 should be brought forward and be more focused; one option perhaps is to merge the text in current section 4 with the corresponding materials in subsections 3.3 – 3.6.
The study otherwise is very clear.
To sum up, I do think this paper has discussed something nice and thus should be accepted for publication; however, the authors should present and discuss all the findings properly.
Author Response
第 1 点:结果和发现应该以更便于读者阅读的方式呈现。我建议将第 3.2 小节中的大表作为附录放在最后。
回复1:感谢您的建议。3.2 小节中的表格已作为表 A1 移至附录文件。
第2点:第4节的讨论应该提出来,更有针对性;一种选择可能是将当前第 4 节中的文本与第 3.3 至 3.6 小节中的相应材料合并。
回应 2:感谢您提出这个极好的建议。我们在第 265-268 行添加了对第 3.3 小节的讨论,我们在最初的提交中省略了。此外,我们对第 4 节进行了重组,将 3.4 和 3.6 的讨论提前,并将机制讨论置于最后。最后,我们更改了第 4 节中的一些表达方式,以将材料纳入第 3.4 至 3.6 小节。

Reviewer 2 Report
Thank you so much for this valuable work. This is an important contribution to the literature and you gave us much to think about in the future. I have only a few comments to improve the manuscript.
1. The analysis is based on age group, duration of intervention, and disease groups. But the background does not touch on any of the three issues and provides a background for the coming analyses. Please expand the background to reflect on these important variables.
2. Figure 1 Header says trail instead of trial. Also, the first box says n=3 it should say 1206 to reflect all included studies or you need to add a box above.
3. The sentence "Nineteen studies were of young adults 149 aged under 45 years old, 20 involved middle-aged adults aged 45-59 years, and 18 studies 150 were of older adults aged 60 years and older." Is worded awkwardly and would flow better if it would be rephrased. Such as "Nineteen studies included younger adults under 45 years old, 20 studies included middle-aged adults between 45-59 years, and 18 studies included older adults above 60 years.
4. What determined the age-group categories?
5. Line 163 Please spell out PSG the first time
6. Table 1 would benefit not only from the author but also from the country where the study was conducted.
7. Line 176 please insert the word "the" after between" there was no heterogeneity in baseline sleep indicators between two groups"
8. Figure 4 I cannot read in its current form and it is unclear what the numbers along the x-axis mean.
9. Figure 5 I am also not able to read.
10. Line 245 Please change the word old for older
Author Response
Point 1: The analysis is based on age group, duration of intervention, and disease groups. But the background does not touch on any of the three issues and provides a background for the coming analyses. Please expand the background to reflect on these important variables.
Response 1: We are very grateful to the reviewer for this suggestion. We add in the introduction (lines 69-80) the rationale for why we perform moderator analysis on these important variables.
Point 2: Figure 1 Header says trail instead of trial. Also, the first box says n=3 it should say 1206 to reflect all included studies or you need to add a box above.
Response 2: Thank you for your careful review. We overlooked the numerical error in Figure 1 and the misspelling of "trial" when we checked the original manuscript. We have revised them in the manuscript.
Point 3: The sentence "Nineteen studies were of young adults 149 aged under 45 years old, 20 involved middle-aged adults aged 45-59 years, and 18 studies 150 were of older adults aged 60 years and older." Is worded awkwardly and would flow better if it would be rephrased. Such as "Nineteen studies included younger adults under 45 years old, 20 studies included middle-aged adults between 45-59 years, and 18 studies included older adults above 60 years.
Response 3: Thank you for pointing this out and providing a better way of rewriting it. We have reorganized the sentences in the manuscript as you suggested.
Point 4: What determined the age-group categories?
Response 4: Thank you for pointing it out. When we determined the age-group categories, we used the criteria used in some studies, middle-aged adults (45-59 years) (Solhi et al., 2022) and older adults (60+ years)(Klimczuk, 2016; McGinnis, 2018). In addition, based on the number of studies included in this paper, we included studies under 40 years of age in another group to make the number of studies in the three groups approximately equal (Younger adults: n=19; Middle-aged adults: n=20; Older adults: n=18).
Point 5: Line 163 Please spell out PSG the first time
Response 5: We would like to thank you for catching this error, and we have added the full term at its first mention in the manuscript.
Point 6: Table 1 would benefit not only from the author but also from the country where the study was conducted.
Response 6: Thank you so much for your excellent suggestion. We agree that the addition of countries where the study was conducted is necessary. We have added a column in table1 in the Appendix to describe this.
Point 7: Line 176 please insert the word "the" after between" there was no heterogeneity in baseline sleep indicators between two groups"
Response 7: We have inserted the word "the" in that sentence as suggested.
Point 8: Figure 4 I cannot read in its current form and it is unclear what the numbers along the x-axis mean.
Response 8: Thank you for pointing out this omission. We have added a note below Figure 4 in the manuscript (line 204).
Point 9: Figure 5 I am also not able to read.
Response 9: Thank you for pointing this out. Figure 5 is from the same direct output version of Stata16.0 as Figure 4. The difference is that there are some hints at the top of it, and we have enlarged this image in the manuscript to make it easier to review.
Point 10: Line 245 Please change the word old for older
Response 10: We agree that a correction to this word is necessary, and thank you for pointing out this error. We have modified the word in the manuscript.
References
Klimczuk, A. (2016). Adulthood. A. Klimczuk, Adulthood,[in:] HL Miller (ed.), The SAGE Encyclopedia of Theory in Psychology, Sage, Thousand Oaks, 2016, 15-18.
McGinnis, D. (2018). Resilience, Life Events, and Well-Being During Midlife: Examining Resilience Subgroups. J Adult Dev, 25(3), 198-221. https://doi.org/10.1007/s10804-018-9288-y
Solhi, M., Pirouzeh, R., & Zanjari, N. (2022). Middle-aged preparation for healthy aging: a qualitative study. BMC Public Health, 22(1), 1-8.
